# Photoinitiated Multicomponent Anti-Markovnikov Alkoxylation over Graphene Oxide

**DOI:** 10.3390/molecules27020475

**Published:** 2022-01-12

**Authors:** Liang Nie, Xiangjun Peng, Haiping He, Jian Hu, Zhiyang Yao, Linyi Zhou, Ming Yang, Fan Li, Qing Huang, Liangxian Liu

**Affiliations:** 1Key Laboratory of Organo-Pharmaceutical Chemistry of Jiangxi Province, Gannan Normal University, Ganzhou 341000, China; a15770740895@163.com (L.N.); zly19136753010@163.com (L.Z.); mmyangyangw@163.com (M.Y.); lf15185987556@163.com (F.L.); 2Key Laboratory of Prevention and Treatment of Cardiovascular and Cerebrovascular Diseases, Ministry of Education, Gannan Medical University, Ganzhou 341000, China; hehaiping223@163.com (H.H.); h1941562258@163.com (J.H.); yaozhiyang0102@163.com (Z.Y.)

**Keywords:** graphene oxide, anti-Markovnikov addition, quinoxalone, alkoxyl radical

## Abstract

The development of graphene oxide–based heterogeneous materials with an economical and environmentally–friendly manner has the potential to facilitate many important organic transformations but proves to have few relevant reported reactions. Herein, we explore the synergistic role of catalytic systems driven by graphene oxide and visible light that form nucleophilic alkoxyl radical intermediates, which enable an anti-Markovnikov addition exclusively to the terminal alkenes, and then the produced benzyl radicals are subsequently added with *N*–methylquinoxalones. This photoinduced cascade radical difunctionalization of olefins offers a concise and applicable protocol for constructing alkoxyl–substituted *N*–methylquinoxalones.

## 1. Introduction

The graphene oxide (GO), as one of the popular carbocatalysts with two-dimensional honeycomb structures, has giant π-conjugated systems, unpaired electrons and several oxygen-containing function groups, such as hydroxy, epoxide, carbonyl and carboxyl groups, which have the acidic, nucleophilic and oxidized capabilities [1,2,3,4,5,6,7,8,9,10]. These unique chemical structures of GOs play an important role in its acidic, nucleophilic and oxidized capabilities. The superior electrical conductivity, high specific surface area and excellent optical transmittance impart to GO crucial properties such as: a template for anchor active species to metal or photocatalysts, and synergistic interactions between them resulting in improved yield [11,12,13,14]. Given such a situation, GO-based photocatalysts offer prospective applications to associate with photoenergy conversion, and the exploration of different approaches in organic transformations have been also improved by GO-based photocatalytic redox processes.

The alkoxyl radicals are versatile reactive intermediates which have been widely exploited in various organic transformations [15,16,17,18,19,20,21]. Compared to aryloxy radicals, this electrophilic O-centered radicals lack the stabilization of mesomeric effects and spin density delocalization compared with aryloxy radicals. The cleavage of N-O bond from various *N*-alkoxypyridinium ions (NAPs) generates the corresponding alkoxyl radicals, affording attractive approaches to the construction of C-O bond under visible light-induced conditions [22,23,24,25]. A plethora of these O-radical intermediates is commonly considered as a powerful toolbox for synthetic and mechanistic studies, which participates in oxidation, hydrogen atom transfer (HAT) and radical addition reactions (Figure 1a). For example, the photogeneration of alkoxy radicals via reductive cleavage of 4-cyano-substituted *N*-methoxypyridinium had oxidated alcohol to yield oxidation product [26]. Recently, the groups of Hong and Zhu independently described a generation of oxygen-centered radicals mediated by NAPs, and subsequent radical translocation easily activated the remote unactivated C(sp^3^)-H bonds by intramolecular 1,5-hydrogen atom transfer (1,5-HAT) [27,28,29]. For no hydrogen abstracted in intermolecular process, alkoxy radicals was efficiently trapped with styrenes by anti-Markovnikov regioselectivity [30,31]. However, the development of multicomponent reactions (MCR) of anti-Markovnikov alkoxylation is still great a challenge without traditional transition–metal catalysts. Thus, we intend to use NAPs that release alkoxyl radical through the synergistic effect of GO and visible-light excitation, and subsequent MCR the approach in anti-Markovnikov addition will have been smooth for constructing C3-substituted alkoxyquinoxalones (Figure 1b).

## 2. Results

The GO material used in this investigation was prepared by Hummers oxidation of graphite and subsequent exfoliation, as reported [32]. The obtained GO material has been characterized by X-ray photoelectron spectroscopy (XPS), scanning electron microscope (SEM), and infrared spectrum (IR) (see the Appendix A).

To validate the above design, we selected *N*-methyl quinoxalone (**1a**), styrene (**2a**) and 4-cyano-substituted *N*-methoxypyridinium salt (**3a**) as pilot substrates to test the difunctional reaction (Table 1). Based on extensive screening of conditions (see Appendix A), we found that when using 80 wt% GO, 0.2 mmol **1a**, 1.5 equivalents of **2a**, and 3 equivalents of **3a** in a mixed solvent of MeCN and H_2_O (70:1 *v*/*v*) at room temperature under the irradiation of 15 W blue LEDs, the expected three-component product **4a** was obtained in 70% yield (Table 1, entry 1). When the other *N*-methoxypyridiniums without substitutes or bearing electron-donating and electron–withdrawing groups were used instead of *N*-methoxypyridinium salt (**3a**), the yields were substantially showed as disappointing results (entries 2–3). Next, the effect of photocatalysts, such as Ru(bpy)_3_Cl_2_ (entry 4) and *fac*-Ir(ppy)_3_, were evaluated, affording 39% and 37% conversion, respectively (entry 5). Notably, the low conversions were observed by using other solvents (entries 6–7). Moreover, adding small amount of water to CH_3_CN had a remarkable impact on the anti-Markovnikov alkoxylation yield (entry 8). The GO loadings were evaluated, the yield of alkoxyquinoxalone **4a** decreased when the GO loadings were 50 wt% or 100 wt% (entries 9–10). Finally, no reaction was observed in the absence of GO as a photocatalyst or light (entry 11).

Encouraged by these promising results, we studied the general applicability of the developed reaction methods using various substituted quinoxalones **1**, styrene **2a**, and 4-cyano-substituted *N*-methoxypyridinium salts **3** to obtain methoxylquinoxalones **4**, and the results are summarized in Figure 2. At first, different *N*-methoxypyridinium salts (BF_4_^−^, OTf^−^, MeSO_4_^−^) as a methoxy source were tested under a nitrogen atmosphere and 15 W blue LEDs irradiation for 24 h (entries 1–3). However, the yields of the desired product were decreased in *N*-methoxypyridinium salts (X = OTf, MeSO_4_) (entries 2 and 3). Thus, *N*-methoxypyridinium salt **3a** was selected as a methoxy source. Subsequently, the *N*-methylquinoxalones moiety bearing electron–rich (6,7-Me) and electron−deficient (6-Cl) were suitable for this transformation, affording the corresponding products **4b** and **4c** in 58% and 55% yield, respectively.

Further, substrates having various *N*-substituents, including benzyl, naphthalen-2-ylmethyl, hexyl, cyclohexylmethyl, allyl, propargyl, and 2-ethoxy-2-oxoethyl groups, reacted with equal ease with **2a** and **3a** to provide the corresponding methoxylquinoxalone derivatives (**4d**–**4n**) in moderate to good yields. Among them, the substituent effect on the benzyl ring was investigated. The results have shown that electron–donating substituents showed better results than electron-withdrawing substituents in this transformation. For example, 4-methylbenzyl derivative afforded the desired **4e** in 60% yield. However, *N*-substituted benzyl derivatives with electron-withdrawing substituents (F, Cl, and CF_3_) provided the desired products **4f**–**4h** in yields ranging from 33% to 45%. Quinoxalone bearing electron-neutral (N-H) was also found to furnish the desired product **4o** in 58% yield. Furthermore, the structure of compound **4f** was unambiguously confirmed by X-ray crystallographic analysis (see Appendix A).

In order to expand the molecular library of methoxylquinoxalone frameworks, our work was extended to the usage of other substituted styrenes, and the results are summarized in Figure 3. These results indicated that the electronic properties of the substituents on these styrenes showed little influence on the efficiency of this reaction. In general, the reactions of styrenes with electron-neutral (4-H), electron−deficient (4-CF_3_, 4-OCOMe), electron-rich (4-Me, 4-Bu-*t*), and halogenated (4-Cl, 3-Br) groups were all compatible to afford the corresponding products in moderate to good yields (43–70%; **4a**, **4p**−**4u**) under the optimal reaction conditions. It is noticed that 2-chlorostyrene was not good alkene substrate presumably due to the sterically hindered (2-Cl) substituent. Similarly, reaction of *N*-methylquinoxalone **1a** and *N*-methoxypyridinium salt **3a** with rigid 2-vinylnaphthalene gave desired product **4y** in 26% yield. In addition, the feasibility of this protocol was further expanded by utilizing *N*-ethoxypyridinium salt **3b** as an ethoxy source, which proceeded successfully under the optimized conditions to give the product **4z** in 62% yield.

UV–vis absorption experiments were performed to examine the role of photocatalysts, and the results showed that GO exhibited evident absorption in the visible-light region (Figure 1a). Based on the absorption experiments, we anticipated no photoactive electron donor–acceptor (EDA) aggregation between GO, quinoxalone **1a**, styrene **2a**, and 4–cyano–substituted *N*–methoxypyridinium **3a**. The fluorescence excitation spectrum were excited at 419 nm and 437 nm (excitation maximum of GO), respectively (Figure 1b). The two values were in the range of 380–500 nm, and so the blue light LED lamp was effective for this reaction. Fluorescence quenching techniques and the Stern-Volmer analysis on **1a**, **2a** and **3a** showed effective quenching of the photoluminescence of GO, which revealed that the quenching of GO was directly proportional to the concentrations of substrates (Figure 1c,d). These investigations showed the single–electron transfer (SET) between GO and substrates under visible–light irradiation.

Various control experiments were conducted to shine light on the plausible mechanism of this three-component cascade process. As showed in Figure 4 (1), no reaction took place, or it was not effective for this anti-Markovnikov addition under an oxygen or air atmosphere. When dibutylhydroxytoluene (BHT) or (2,2,6,6-tetramethylpiperidin-1-yl)oxyl (TEMPO) was put into the standard reaction conditions, the reaction was completely suppressed, strongly suggesting the involvement of alkoxy radicals in product formation (Figure 4 (2) and (3)). Moreover, the radical addition products **I** and **II** were detected by LC-MS, indicating the presence of methoxybenzyl radicals **A** after methoxy radical intermediates addition to the styrenes (Figure 4 (3)). In addition, the radical clock reaction of diethyl 2,2-diallylmalonate **5** with **1a** and **3a** delivered the cyclopentane derivative **III** in favor of the cis-selectivity assigned on the Beckwith-Houk model (Figure 4 (4)) [33,34,35]. These results indicated that free radical mechanism for this three-component cascade transformation was initiated by the alkoxy radical.

The catalyst reusability was examined. The GO was separated by filtration after the first reaction run and used for the second one under the same conditions. However, the yield of **3a** is very low, indicating that the recovered-GO is ineffective.

X-ray photoelectron spectroscopy (XPS) and Fourier transform infrared (FT-IR) analysis showed that a large of epoxide groups and carbonyl functional groups are lost when GO participates the reaction (Appendix A). The GO and recovered-GO (GO-R) were analyzed by XPS in order to study the surface chemical state and chemical composition of GO and GO-R. The full-scale XPS spectrum (Appendix A) proved that GO have C, O and S elements. For GO-R, the presence of C, O, S, F and B were confirmed by survey XPS spectra, indicating that the anion BF_4_^−^ was successfully doped in carbon catalyst.

Based on the current results and previous reports [36,37], the following tentative SET mechanism for this metal–free domino process was depicted in Figure 5. Initiation proceeds had driven by the synergistic effect of GO and visible light-mediated C-O bond cleavage of *N*-methoxypyridinium that generated the corresponding methoxy-radical. Then, methoxy radical added to the C=C bond of styrene **2a**, delivering nucleophilic *β*-methoxylated radical **A**. Sequentially, the addition of chemo-selective intermediate **A** to the electrophilic C=N bond in substrate **1a** afforded nitrogen radical intermediate **B**, which could be further oxidized by GO through SET and deprotonation process to give the desired product **4a**.

## 3. Materials and Methods

### 3.1. General Information

Unless otherwise specified, commercial reagents and solvents were used without further purification. Commercially available chemicals were purchased from Leyan (Shanghai Haohong Scientific Co., Ltd. Shanghai, China) and used without any further purification. ^1^H and ^13^C NMR spectra were recorded on a Bruker spectrometers at 400 and 100 MHz, respectively. The chemical shifts were given in parts per million, relative to CDCl_3_ (7.26 ppm for ^1^H) and CDCl_3_ (77.0 ppm for ^13^C. Peak multiplicities were reported as follows: s, singlet; d, doublet; t, triplet; m, multiplet; br. s, broad singlet and *J*, coupling constant (Hz). Mass spectra were recorded with Bruker Dalton Esquire 3000 plus LC–MS apparatus. Elemental analyses were carried out on a Perkin-Elmer 240B instrument. HRFABMS spectra were recorded on a FTMS apparatus. Silica gel (300–400 mesh) was used for flash column chromatography, eluting (unless otherwise stated) with an ethyl acetate/petroleum ether (PE) (60–90 °C) mixture.

### 3.2. General Procedure of the Products **4**

To a mixed solvent of MeCN and H_2_O (70:1 *v*/*v*) of quinoxalone **1** (0.2 mmol), *N*−methoxypyridinium salt **3a** (0.6 mmol), and GO (80 wt%) was added styrene **2a** (0.3 mmol) under an argon atmosphere irradiated by 15 w blue LEDs and the mixture was stirred at room temperature for 24 h. The reaction mixture was concentrated under reduced pressure. The residue was purified by flash chromatography on silica gel (eluent: EtOAc/PE = 1:4) to yield the corresponding product **4**.

### 3.3. Characterization Data of Products **4**

3-(2-Methoxyl-1-phenylethyl)-methylquinoxalin-2(1*H*)-one **4a**. Yellow solid (41 mg, 70%). Mp: 98−101 °C. ^1^H NMR (400 MHz, CDCl_3_): *δ* 7.96 (dd, *J* = 8.0, 1.2 Hz, 1H), 7.53 (dt, *J* = 1.2, 8.5 Hz, 1H), 7.50 (d, *J* = 8.5 Hz, 2H), 7.36 (t, *J* = 7.3 Hz, 1H), 7.31 (t, *J* = 7.3 Hz, 2H), 7.27 (d, *J* = 8.0 Hz, 1H), 7.23 (t, *J* = 7.3 Hz, 1H), 5.06 (dd, *J* = 9.0, 6.3 Hz, 1H), 4.41 (t, *J* = 9.0 Hz, 1H), 3.92 (dd, *J* = 9.0, 6.3 Hz, 1H), 3.64 (s, 3H), 3.41 (s, 3H). ^13^C NMR (100 MHz, CDCl_3_): *δ* 159.3, 154.6, 138.7, 133.1, 132.7, 130.3, 129.9, 128.7, 128.5, 127.1, 123.5, 113.5, 74.8, 59.0, 47.3, 29.1. HRESIMS calcd for [C_18_H_18_N_2_O_2_ + H]^+^ 295.1447, found 295.1454.

3-(2-Methoxy-1-phenylethyl)-1,6,7-trimethylquinoxalin-2(1*H*)-one **4b**. Yellow solid (37 mg, 58%), Mp: 129–132 °C. ^1^H NMR (400 MHz, CDCl_3_): *δ* 7.71 (s, 1H), 7.49 (d, *J* = 7.4 Hz, 2H), 7.30 (t, *J* = 7.4 Hz, 2H), 7.21 (t, *J* = 7.4 Hz, 1H), 7.03 (s, 1H), 5.04 (t, *J* = 7.1 Hz, 1H), 4.37 (t, *J* = 8.9 Hz, 1H), 3.92 (t, *J* = 7.9 Hz, 1H), 3.61 (s, 3H), 3.40 (s, 3H), 2.42 (s, 3H), 2.38 (s, 3H). ^13^C NMR (100 MHz, CDCl_3_): *δ* 157.9, 154.6, 139.7, 139.0, 132.3, 131.1, 130.3, 128.7, 128.5, 128.4, 127.0, 114.1, 74.9, 59.0, 47.1, 29.1, 20.6, 19.2. HRESIMS calcd for [C_20_H_22_N_2_O_2_ + H]^+^ 323.1760, found 323.1769.

6-Chloro-3-(2-methoxyl-1-phenylethyl)-1-methylquinoxalin-2(1*H*)-one **4c**. Yellow solid (36 mg, 55%). Mp: 127–130 °C. ^1^H NMR (400 MHz, CDCl_3_): *δ* 7.96 (s, 1H), 7.48 (d, *J* = 8.5 Hz, 2H), 7.47 (s, 1H), 7.31 (t, *J* = 7.0 Hz, 2H), 7.24 (d, *J* = 7.0 Hz, 1H), 7.19 (d, *J* = 8.5 Hz, 1H), 5.07 (t, *J* = 6.7 Hz, 1H), 4.38 (t, *J* = 9.1 Hz, 1H), 3.87 (dd, *J* = 6.7, 1.8 Hz, 1H), 3.61 (s, 3H), 3.40 (s, 3H). ^13^C NMR (100 MHz, CDCl_3_): *δ* 160.7, 154.2, 138.2, 133.2, 131.8, 129.9, 129.5, 128.8, 128.7, 128.6, 127.3, 114.7, 74.6, 59.0, 47.4, 29.3. HRESIMS calcd for [C_18_H_17_ClN_2_O_2_ + H]^+^ 329.1057, found 329.1067.

1-Benzyl-3-(2-methoxy-1-phenylethyl)quinoxalin-2(1*H*)-one **4d**. Yellow solid (48 mg, 65%). Mp: 85–87 °C. ^1^H NMR (400 MHz, CDCl_3_): *δ* 7.99 (d, *J* = 7.8 Hz, 1H), 7.55 (d, *J* = 7.4 Hz, 2H), 7.41 (t, *J* = 7.7 Hz, 1H), 7.38–7.27 (m, 8H), 7.20 (d, *J* = 7.4 Hz, 2H), 5.54 (d, *J* = 15.6 Hz, 1H), 5.34 (d, *J* = 15.6 Hz, 1H), 5.17 (t, *J* = 7.0 Hz, 1H), 4.46 (t, *J* = 8.9 Hz, 1H), 3.98 (t, *J* = 8.3 Hz, 1H), 3.45 (s, 3H). ^13^C NMR (100 MHz, CDCl_3_): *δ* 159.4, 154.7, 138.7, 135.3, 133.0, 132.5, 130.4, 129.9, 128.9, 128.8, 128.6, 127.6, 127.2, 126.9, 123.6, 114.4. 74.9, 59.1, 47.4, 46.0. HRESIMS calcd for [C_24_H_23_N_2_O_4_ + H]^+^ 371.1754, found 371.1769.

3-(2-Methoxyl-1-phenylethyl)-1-(4-methylbenzyl)quinoxalin-2(1*H*)-one **4e**. Yellow oil (46 mg, 60%). ^1^H NMR (400 MHz, CDCl_3_): *δ* 7.95 (d, *J* = 7.9 Hz, 1H), 7.52 (d, *J* = 7.5 Hz, 2H), 7.42 (t, *J* = 7.5 Hz, 1H), 7.33 (t, *J* = 7.1 Hz, 2H), 7.31 (t, *J* = 7.1 Hz, 1H), 7.25 (d, *J* = 7.9 Hz, 2H), 7.10 (s, 4H), 5.50 (d, *J* = 15.5 Hz, 1H), 5.29 (d, *J* = 15.5 Hz, 1H), 5.13 (t, *J* = 6.5 Hz, 1H), 4.43 (t, *J* = 9.1 Hz, 1H), 3.95 (dd, *J* = 9.1, 6.5 Hz, 1H), 3.43 (s, 3H), 2.31 (s, 3H). ^13^C NMR (100 MHz, CDCl_3_): *δ* 159.4, 154.6, 138.7, 137.3, 132.9, 132.5, 132.3, 130.4, 129.9, 129.5, 128.7, 128.5, 127.1, 126.9, 123.5, 114.4, 74.9, 59.0, 47.4, 45.8, 21.1. HRESIMS calcd for [C_25_H_24_N_2_O_2_ + H]^+^ 385.1916, found 385.1929.

1-(4-Fluorobenzyl)-3-(2-methoxy-1-phenylethyl)quinoxalin-2(1*H*)-one **4f**. White solid (27 mg, 35%). Mp: 114–116 °C. ^1^H NMR (400 MHz, CDCl_3_): *δ* 7.94 (d, *J* = 7.4 Hz, 1H), 7.48 (d, *J* = 7.2 Hz, 2H), 7.40 (t, *J* = 7.2 Hz, 1H), 7.38–7.13 (m, 7H), 6.94 (d, *J* = 8.4 Hz, 2H), 5.44 (d, *J* = 15.3 Hz, 1H), 5.27 (d, *J* = 15.3 Hz, 1H), 5.09 (t, *J* = 7.3 Hz, 1H), 4.40 (d, *J* = 9.0 Hz, 1H), 3.91 (dd, *J* = 9.0, 7.3 Hz, 1H), 3.39 (s, 3H). ^13^C NMR (100 MHz, CDCl_3_): *δ* 162.0 (d, *J* = 246.4 Hz), 159.4, 154.5, 138.6, 132.9, 132.3, 131.0 (d, *J* = 3.1 Hz), 130.5, 129.9, 128.8, 128.7, 128.5, 127.1, 123.6, 115.8 (d, *J* = 21.7 Hz), 114.1, 74.8, 59.0, 47.4, 45.3. ^19^F NMR (376 MHz, CDCl_3_): *δ* −114.52; HRESIMS calcd for [C_24_H_21_FN_2_O_2_ + H]^+^ 389.1665, found 389.1651.

1-(2,6-Dichlorobenzyl)-3-(2-methoxy-1-phenylethyl)quinoxalin-2(1*H*)-one **4g**. Yellow solid (29 mg, 33%). Mp: 85–86 °C. ^1^H NMR (400 MHz, CDCl_3_): *δ* 8.10 (d, *J* = 8.0 Hz, 1H), 7.87 (d, *J* = 8.0 Hz, 1H), 7.65 (t, *J* = 7.3 Hz, 1H), 7.59 (t, *J* = 7.3 Hz, 1H), 7.40 (d, *J* = 7.6 Hz, 2H), 7.30 (t, *J* = 7.6 Hz, 1H), 7.26 (d, *J* = 7.6 Hz, 2H), 7.18 (s, 3H), 5.79 (d, *J* = 11.4 Hz, 1H), 5.64 (d, *J* = 11.4 Hz, 1H), 4.80 (t, *J* = 7.0 Hz, 1H), 4.48 (t, *J* = 8.5 Hz, 1H), 4.02 (t, *J* = 7.9 Hz, 1H), 3.39 (s, 3H). ^13^C NMR (100 MHz, CDCl_3_): *δ* 155.2, 149.7, 139.7, 139.0, 138.8, 137.2, 132.1, 130.5, 129.3, 128.9, 128.7, 128.4, 128.3, 126.9, 126.8, 126.4, 74.6, 63.2, 59.1, 47.4. HRESIMS calcd for [C_24_H_20_Cl_2_N_2_O_2_ + H]^+^ 439.0980, found 439.0989.

3-(2-Methoxyl-1-phenylethyl)-1-(4-(trifluoromethyl)benzyl)quinoxalin-2(1*H*)-one **4h**. Yellow oil (39 mg, 45%). ^1^H NMR (400 MHz, CDCl_3_): *δ* 8.00 (d, *J* = 7.9 Hz, 1H), 7.55 (d, *J* = 7.9 Hz, 2H), 7.52 (d, *J* = 7.9 Hz, 2H), 7.44 (t, *J* = 7.7 Hz, 1H), 7.36 (t, *J* = 7.7 Hz, 2H), 7.33 (t, *J* = 7.7 Hz, 1H), 7.31 (t, *J* = 7.5 Hz, 2H), 7.26 (t, *J* = 7.5 Hz, 1H), 7.14 (d, *J* = 7.9 Hz, 1H), 5.56 (d, *J* = 15.9 Hz, 1H), 5.40 (d, *J* = 15.9 Hz, 1H), 5.12 (t, *J* = 7.3 Hz, 1H), 4.45 (t, *J* = 9.1 Hz, 1H), 3.94 (dt, *J* = 9.1, 7.3 Hz, 1H), 3.44 (s, 3H). ^13^C NMR (100 MHz, CDCl_3_): *δ* 159.4, 154.5, 139.3, 138.5, 132.9, 132.2, 130.6, 130.1, 129.8, 128.7, 128.6, 127.2, 127.1, 125.9 (d, *J* = 7.7 Hz), 123.9 (q, *J* = 272.1 Hz), 123.8, 113.9, 74.8, 59.1, 47.5, 45.6. ^19^F NMR (376 MHz, CDCl_3_) *δ* −62.64; HRESIMS calcd for [C_25_H_21_F_3_N_2_O_2_ + H]^+^ 439.1633, found 439.1621.

3-(2-Methoxy-1-phenylethyl)-1-(naphthalen-2-ylmethyl)quinoxalin-2(1*H*)-one **4i**. Yellow solid (29 mg, 35%). Mp: 56–58 °C. ^1^H NMR (400 MHz, CDCl_3_): *δ* 7.99 (d, *J* = 8.0 Hz, 1H), 7.80 (d, *J* = 4.9 Hz, 2H), 7.72 (t, *J* = 4.9 Hz, 1H), 7.59 (d, *J* = 7.9 Hz, 2H), 7.56 (s, 1H), 7.47 (t, *J* = 4.9 Hz, 2H), 7.40–7.30 (m, 7H), 5.72 (d, *J* = 15.8 Hz, 1H), 5.48 (d, *J* = 15.8 Hz, 1H), 5.19 (t, *J* = 7.3 Hz, 1H), 4.49 (t, *J* = 9.0 Hz, 1H), 3.99 (t, *J* = 7.8 Hz, 1H), 3.47 (s, 3H). ^13^C NMR (100 MHz, CDCl_3_): *δ* 159.4, 154.7, 138.7, 133.3, 133.0, 132.8, 132.7, 132.5, 130.4, 130.0, 128.9, 128.8, 128.6, 127.8, 127.7, 127.2, 126.4, 126.1, 125.6, 124.8, 123.6, 114.4, 74.9, 59.1, 47.5, 46.3. HRESIMS calcd for [C_28_H_24_N_2_O_2_ + H]^+^ 421.1916, found 421.1933.

1-Hexyl-3-(2-methoxy-1-phenylethyl)quinoxalin-2(1*H*)-one **4j**. Yellow oli (35 mg, 48%). ^1^H NMR (400 MHz, CDCl_3_): *δ* 7.97 (d, *J* = 8.0 Hz, 1H), 7.52 (t, *J* = 7.0 Hz, 2H), 7.51 (s, 1H), 7.36–7.21 (m, 5H), 5.10 (t, *J* = 7.6 Hz, 1H), 4.42 (t, *J* = 9.0 Hz, 1H), 4.24 (dt, *J* = 7.8, 14.3 Hz, 1H), 4.11 (dt, *J* = 7.8, 14.3 Hz, 1H), 3.94 (t, *J* = 7.6 Hz, 1H), 3.42 (s, 3H), 1.71 (dt, *J* = 6.8, 15.5 Hz, 1H), 1.45–1.28 (m, 6H), 0.92 (t, *J* = 6.0 Hz, 1H). ^13^C NMR (100 MHz, CDCl_3_): *δ* 159.2, 154.2, 138.8, 133.0, 132.3, 130.5, 129.8, 128.7, 128.5, 127.1, 123.2, 113.5, 74.8, 59.0, 47.2, 42.5, 31.5, 27.2, 26.7, 22.6, 14.0. HRESIMS calcd for [C_23_H_28_N_2_O_2_ + H]^+^ 365.2229, found 365.2211.

1-(Cyclohexylmethyl)-3-(2-methoxy-1-phenylethyl)quinoxalin-2(1*H*)-one **4k**. Colorless oil (48 mg, 64%). ^1^H NMR (400 MHz, CDCl_3_): *δ* 8.07 (d, *J* = 7.9 Hz, 1H), 7.78 (d, *J* = 7.9 Hz, 1H), 7.61 (t, *J* = 7.5 Hz, 1H), 7.55 (t, *J* = 7.1 Hz, 1H), 7.38 (d, *J* = 7.1 Hz, 2H), 7.28 (t, *J* = 6.8 Hz, 2H), 7.22 (d, *J* = 6.8 Hz, 1H), 4.85 (t, *J* = 6.4 Hz, 1H), 4.48 (t, *J* = 8.3 Hz, 1H), 4.22 (dd, *J* = 18.6, 5.6 Hz, 2H), 4.01 (t, *J* = 8.3 Hz, 1H), 3.41 (s, 3H), 1.82–1.68 (m, 7H), 1.30–1.23 (m, 2H), 1.08–1.00 (m, 2H). ^13^C NMR (100 MHz, CDCl_3_): *δ* 156.0, 149.8, 139.9, 139.4, 138.4, 129.1, 128.8, 128.7, 128.3, 126.9, 126.6, 126.0, 75.1, 71.6, 59.1, 47.5, 37.4, 29.8, 26.5, 25.8. HRESIMS calcd for [C_24_H_28_N_2_O_2_ + H]^+^ 377.2229, found 377.2254.

Ethyl 2-(3-(2-methoxy-1-phenylethyl)-2-oxoquinoxalin-1(2*H*)-yl)acetate **4l**. White solid (40 mg, 60%). Mp: 86–89 °C. ^1^H NMR (400 MHz, CDCl_3_): *δ* 7.88 (dd, *J* = 8.0, 1.4 Hz, 1H), 7.41 (dt, *J* = 2.0, 8.0 Hz, 1H), 7.39 (dd, *J* = 3.6, 2.0 Hz, 1H), 7.37 (s, 1H), 7.28 (dt, *J* = 1.4, 8.0 Hz, 1H), 7.21 (t, *J* = 7.5 Hz, 2H), 7.13 (dt, *J* = 2.0, 7.5 Hz, 1H), 6.95 (d, *J* = 8.0 Hz, 1H), 4.95 (d, *J* = 17.3 Hz, 1H), 4.94 (dd, *J* = 8.3, 6.6 Hz, 1H), 4.78 (d, *J* = 17.3 Hz, 1H), 4.28 (dd, *J* = 9.3, 8.6 Hz, 1H), 4.11 (m, 2H), 3.83 (dd, *J* = 9.3, 6.6 Hz, 1H), 3.31 (s, 3H), 1.14 (t, *J* = 7.1 Hz, 3H). ^13^C NMR (100 MHz, CDCl_3_): *δ* 167.1, 159.0, 154.1, 138.5, 132.7, 132.2, 130.6, 130.1, 128.7, 128.6, 127.2, 123.8, 113.1, 74.7, 62.0, 59.0, 47.3, 43.7, 14.1. HRESIMS calcd for [C_21_H_22_N_2_O_4_ + H]^+^ 367.1658, found 367.1682.

1-Allyl-3-(2-methoxy-1-phenylethyl)quinoxalin-2(1*H*)-one **4m**. White solid (38 mg, 60%). Mp: 76–78 °C. ^1^H NMR (400 MHz, CDCl_3_): *δ* 7.98 (d, *J* = 7.9 Hz, 1H), 7.53 (d, *J* = 7.5 Hz, 2H), 7.49 (t, *J* = 7.9 Hz, 1H), 7.35 (t, *J* = 7.9 Hz, 1H), 7.33 (t, *J* = 7.5 Hz, 2H), 7.26 (t, *J* = 8.3 Hz, 1H), 7.24 (t, *J* = 8.3 Hz, 1H), 5.95–5.83 (m, 1H), 5.24 (d, *J* = 10.3 Hz, 1H), 5.16 (d, *J* = 17.9 Hz, 1H), 5.11 (t, *J* = 7.1 Hz, 1H), 4.92 (dd, *J* = 15.9, 4.3 Hz, 1H), 4.77 (dd, *J* = 15.9, 4.3 Hz, 1H), 4.43 (t, *J* = 9.0 Hz, 1H), 3.94 (t, *J* = 7.7 Hz, 1H), 3.43 (s, 3H). ^13^C NMR (100 MHz, CDCl_3_): *δ* 159.3, 154.1, 138.7, 132.9, 132.4, 130.8, 130.4, 129.9, 128.7, 128.5, 127.1, 123.5, 118.2, 114.1, 74.9, 59.0, 47.3, 44.7. HRESIMS calcd for [C_20_H_20_N_2_O_2_ + H]^+^ 321.1603, found 321.1625.

3-(2-Methoxy-1-phenylethyl)-1-(prop-2-yn-1-yl)quinoxalin-2(1*H*)-one **4n**. Yellow solid (38 mg, 60%). Mp: 87–90 °C. ^1^H NMR (400 MHz, CDCl_3_): *δ* 7.97 (d, *J* = 7.6 Hz, 1H), 7.57 (t, *J* = 7.6 Hz, 1H), 7.50 (d, *J* = 7.1 Hz, 2H), 7.44 (d, *J* = 8.3 Hz, 1H), 7.39 (t, *J* = 8.3 Hz, 1H), 7.31 (d, *J* = 7.1 Hz, 2H), 7.24 (t, *J* = 7.1 Hz, 1H), 5.08 (d, *J* = 17.4 Hz, 1H), 5.07 (t, *J* = 6.8 Hz, 1H), 4.90 (d, *J* = 17.4 Hz, 1H), 4.40 (t, *J* = 9.0 Hz, 1H), 3.90 (d, *J* = 7.9 Hz, 1H), 3.41 (s, 3H), 2.27 (s, 1H). ^13^C NMR (100 MHz, CDCl_3_): *δ* 159.1, 153.5, 138.4, 132.9, 131.6, 130.4, 130.0, 128.7, 128.5, 127.2, 123.8, 114.1, 76.8, 74.8, 73.2, 59.0, 47.3, 31.6. HRESIMS calcd for [C_20_H_18_N_2_O_2_ + H]^+^ 319.1447, found 319.1469.

3-(2-Methoxy-1-phenylethyl)quinoxalin-2(1*H*)-one **4o**. Yellow solid (32 mg, 58%). Mp: 160–163 °C. ^1^H NMR (400 MHz, CDCl_3_): *δ* 12.24 (s, 1H), 7.92 (d, *J* = 8.0 Hz, 1H), 7.50 (d, *J* = 7.6 Hz, 1H), 7.49 (t, *J* = 8.5 Hz, 2H), 7.35 (t, *J* = 7.6 Hz, 1H), 7.30 (t, *J* = 8.5 Hz, 2H), 7.22 (t, *J* = 8.0 Hz, 2H), 5.07 (t, *J* = 7.0 Hz, 1H), 4.41 (t, *J* = 8.8 Hz, 1H), 3.95 (t, *J* = 7.6 Hz, 1H), 3.41 (s, 3H). ^13^C NMR (100 MHz, CDCl_3_): *δ* 159.7, 156.2, 138.6, 132.8, 130.9, 129.9, 129.2, 128.7, 128.5, 127.1, 124.0, 115.6, 74.8, 59.1, 46.8. HRESIMS calcd for [C_17_H_16_N_2_O_2_ + H]^+^ 281.1290, found 281.1296.

3-(2-Methoxyl-1-(*p*-tolyl)ethyl)-1-methylquinoxalin-2(1*H*)-one **4p**. White solid (34 mg, 55%). Mp: 107–110 °C. ^1^H NMR (400 MHz, CDCl_3_): *δ* 7.95 (d, *J* = 7.5 Hz, 1H), 7.53 (t, *J* = 8.1 Hz, 1H), 7.40 (d, *J* = 7.7 Hz, 2H), 7.36 (t, *J* = 7.5 Hz, 1H), 7.26 (d, *J* = 8.1 Hz, 1H), 7.13 (d, *J* = 7.7 Hz, 2H), 5.04 (t, *J* = 6.7 Hz, 1H), 4.40 (t, *J* = 9.0 Hz, 1H), 3.92 (dd, *J* = 9.0, 6.7 Hz, 1H), 3.63 (s, 3H), 3.42 (s, 3H), 2.31 (s, 3H). ^13^C NMR (100 MHz, CDCl_3_): *δ* 159.4, 154.5, 136.7, 135.6, 133.1, 132.7, 130.2, 129.9, 129.3, 128.6, 123.4, 113.5, 74.8, 59.0, 46.9, 29.1, 21.1. HRESIMS calcd for [C_19_H_20_N_2_O_2_ + H]^+^ 309.1603, found 309.1021.

3-(1-(4-(*tert*-Butyl)phenyl)-2-methoxyl)-1-methylquinoxalin-2(1*H*)-one **4q**. Yellow oil (32 mg, 55%). ^1^H NMR (400 MHz, CDCl_3_): *δ* 7.95 (dd, *J* = 8.3, 1.1 Hz, 1H), 7.52 (dt, *J* = 1.4, 8.6 Hz, 1H), 7.43 (dt, *J* = 1.4, 8.6 Hz, 2H), 7.36 (dt, *J* = 1.1, 8.3 Hz, 1H), 7.33 (dt, *J* = 1.4, 8.6 Hz, 2H), 7.26 (d, *J* = 8.3 Hz, 1H), 5.07 (dd, *J* = 9.2, 6.0 Hz, 1H), 4.42 (t, *J* = 9.2 Hz, 1H), 3.90 (dd, *J* = 9.2, 6.0 Hz, 1H), 3.64 (s, 3H), 3.41 (s, 3H), 1.29 (s, 9H). ^13^C NMR (100 MHz, CDCl_3_): *δ* 159.4, 154.6, 149.7, 135.4, 133.1, 132.7, 130.2, 129.8, 128.3, 125.5, 123.4, 113.5, 74.8, 59.0, 46.7, 34.4, 31.3, 29.1. HRESIMS calcd for [C_22_H_26_N_2_O_2_ + H]^+^ 351.2073, found 351.2075.

3-(1-(4-Chlorophenyl)-2-methoxyethyl)-1-methylquinoxalin-2(1*H*)-one **4r**. Yellow solid (28 mg, 43%). Mp: 124–126 °C. ^1^H NMR (400 MHz, CDCl_3_): *δ* 7.94 (d, *J* = 8.0 Hz, 1H), 7.55 (d, *J* = 7.8 Hz, 1H), 7.44 (d, *J* = 8.0 Hz, 2H), 7.37 (t, *J* = 7.8 Hz, 1H), 7.28 (s, 1H), 7.27 (d, *J* = 8.0 Hz, 2H), 5.03 (t, *J* = 7.4 Hz, 1H), 4.32 (t, *J* = 8.8 Hz, 1H), 3.92 (t, *J* = 8.1 Hz, 1H), 3.64 (s, 3H), 3.40 (s, 3H). ^13^C NMR (100 MHz, CDCl_3_): *δ* 158.8, 154.4, 137.2, 133.1, 132.9, 132.6, 130.3, 130.2, 130.1, 128.6, 123.6, 113.6, 74.6, 59.0, 46.7, 29.2. HRESIMS calcd for [C_18_H_17_ClN_2_O_2_ + H]^+^ 329.1057, found 329.1077.

3-(2-Methoxyl-1-(4-(trifluoromethyl)phenyl)ethyl)-1-methylquinoxalin-2(1*H*)-one **4s**. White solid (36 mg, 50%). Mp: 118–122 °C. ^1^H NMR (400 MHz, CDCl_3_): *δ* 7.95 (d, *J* = 8.0 Hz, 1H), 7.63 (d, *J* = 8.1 Hz, 2H), 7.57 (t, *J* = 8.0 Hz, 1H), 7.56 (d, *J* = 8.1 Hz, 2H), 7.39 (t, *J* = 7.6 Hz, 1H), 7.30 (d, *J* = 7.6 Hz, 1H), 5.11 (t, *J* = 7.3 Hz, 1H), 4.34 (t, *J* = 8.6 Hz, 1H), 3.98 (t, *J* = 8.6 Hz, 1H), 3.66 (s, 3H), 3.41 (s, 3H). ^13^C NMR (100 MHz, CDCl_3_): *δ* 158.4, 154.4, 142.9, 133.1, 132.6, 130.3, 129.4, 129.1, 128.8, 125.4 (d, *J* = 3.8 Hz), 124.2 (d, *J* = 272.0 Hz), 123.7, 113.6, 74.5, 59.1, 47.2, 29.2. ^19^F NMR (376 MHz, CDCl_3_) *δ* −62.49; HRESIMS calcd for [C_19_H_17_F_3_N_2_O_2_ + H]^+^ 363.1320, found 363.1334.

4-(2-Methoxy-1-(4-methyl-3-oxo-3,4-dihydroquinoxalin-2-yl)ethyl)phenyl acetate **4t**. Yellow solid (37 mg, 52%). Mp: 128–130 °C. ^1^H NMR (400 MHz, CDCl_3_): *δ* 7.91 (dd, *J* = 8.0, 1.4 Hz, 1H), 7.52 (dt, *J* = 1.4, 8.5 Hz, 1H), 7.49 (dt, *J* = 2.6, 7.6 Hz, 2H), 7.34 (dt, *J* = 1.4, 7.6 Hz, 1H), 7.25 (d, *J* = 8.5 Hz, 1H), 6.99 (dt, *J* = 2.6, 8.6 Hz, 2H), 5.06 (dd, *J* = 8.6, 6.3 Hz, 1H), 4.36 (t, *J* = 9.0 Hz, 1H), 3.88 (dd, *J* = 9.0, 6.3 Hz, 1H), 3.63 (s, 3H), 3.38 (s, 3H), 2.26 (s, 3H). ^13^C NMR (100 MHz, CDCl_3_): *δ* 169.6, 158.9, 154.5, 149.7, 136.1, 133.1, 132.6, 130.2, 130.1, 129.8, 123.5, 121.5, 113.6, 74.6, 59.0, 46.6, 29.2, 21.2. HRESIMS calcd for [C_20_H_20_N_2_O_4_ + H]^+^ 353.1501, found 353.1487.

3-(1-(3-Bromophenyl)-2-methoxyethyl)-1-methylquinoxalin-2(1*H*)-one **4u**. White solid (34 mg, 45%). Mp: 96–101 °C. ^1^H NMR (400 MHz, CDCl_3_): *δ* 7.94 (d, *J* = 8.2 Hz, 1H), 7.62 (d, *J* = 1.1 Hz, 1H), 7.53 (t, *J* = 7.8 Hz, 1H), 7.45 (d, *J* = 7.8 Hz, 1H), 7.36 (t, *J* = 7.1 Hz, 1H), 7.35 (d, *J* = 7.1 Hz, 1H), 7.26 (d, *J* = 8.2 Hz, 1H), 7.18 (t, *J* = 7.8 Hz, 1H), 5.02 (t, *J* = 7.7 Hz, 1H), 4.32 (t, *J* = 8.8 Hz, 1H), 3.92 (t, *J* = 8.8, 7.7 Hz, 1H), 3.63 (s, 3H), 3.40 (s, 3H). ^13^C NMR (100 MHz, CDCl_3_): *δ* 158.5, 154.4, 141.1, 133.1, 132.6, 131.5, 130.3, 130.2, 130.1, 130.0, 127.7, 123.6, 122.5, 113.6, 74.6, 59.1, 46.9, 29.2. HRESIMS calcd for [C_18_H_17_BrN_2_O_2_ + H]^+^ 373.0552, found 373.0558.

3-(2-Methoxyl-1-(*o*-tolyl)ethyl)-1-methylquinoxalin-2(1*H*)-one **4v**. Yellow solid (32 mg, 52%). Mp:129–132 °C. ^1^H NMR (400 MHz, CDCl_3_): *δ* 8.00 (d, *J* = 7.8 Hz, 1H), 7.54 (t, *J* = 7.8 Hz, 1H), 7.38 (t, *J* = 7.6 Hz, 1H), 7.29–7.22 (m, 3H), 7.13 (t, *J* = 7.3 Hz, 1H), 7.08 (t, *J* = 7.3 Hz, 1H), 5.30 (dd, *J* = 8.6, 5.7 Hz, 1H), 4.42 (t, *J* = 9.2 Hz, 1H), 3.75 (dd, *J* = 9.2, 5.7 Hz, 1H), 3.63 (s, 3H), 3.42 (s, 3H), 2.72 (s, 3H). ^13^C NMR (100 MHz, CDCl_3_): *δ* 159.6, 154.6, 137.5, 136.9, 133.1, 132.7, 130.7, 130.3, 129.9, 127.1, 126.9, 125.8, 123.4, 113.5, 74.8, 59.1, 42.9, 29.1, 20.0. HRESIMS calcd for [C_19_H_20_N_2_O_2_ + H]^+^ 309.1603, found 309.1612.

3-(1-(2,5-Dimethylphenyl)-2-methoxyethyl)-1-methylquinoxalin-2(1*H*)-one **4w**. Yellow solid (22 mg, 35%). Mp: 58–61 °C. ^1^H NMR (400 MHz, CDCl_3_): *δ* 8.02 (d, *J* = 8.0 Hz, 1H), 7.55 (t, *J* = 7.6 Hz, 1H), 7.40 (t, *J* = 7.6 Hz, 1H), 7.28 (d, *J* = 8.0 Hz, 1H), 7.13 (d, *J* = 7.6 Hz, 1H), 7.04 (s, 1H), 6.95 (d, *J* = 7.6 Hz, 1H), 5.26 (dd, *J* = 8.8, 5.5 Hz, 1H), 4.42 (t, *J* = 9.2 Hz, 1H), 3.75 (dd, *J* = 9.2, 5.5 Hz, 1H), 3.63 (s, 3H), 3.43 (s, 3H), 2.68 (s, 3H), 2.23 (s, 3H). ^13^C NMR (100 MHz, CDCl_3_): *δ* 159.7, 154.6, 136.6, 135.1, 134.3, 133.1, 132.7, 130.6, 130.3, 129.8, 127.8, 127.7, 123.4, 113.5, 74.7, 59.0, 43.0, 29.1, 21.1, 19.6. HRESIMS calcd for [C_20_H_22_N_2_O_2_ + H]^+^ 323.1760, found 323.1781.

3-(1-(2-Chlorophenyl)-2-methoxyethyl)-1-methylquinoxalin-2(1*H*)-one **4x**. Yellow solid (16 mg, 25%). Mp: 94–96 °C. ^1^H NMR (400 MHz, CDCl_3_): *δ* 7.96 (d, *J* = 7.9 Hz, 1H), 7.57 (t, *J* = 7.9 Hz, 1H), 7.44 (d, *J* = 7.5 Hz, 1H), 7.38 (t, *J* = 7.5 Hz, 1H), 7.32 (t, *J* = 8.0 Hz, 1H), 7.31 (d, *J* = 8.0 Hz, 1H), 7.16 (dt, *J* = 4.9, 8.0 Hz, 2H), 5.61 (t, *J* = 7.0 Hz, 1H), 4.33 (t, *J* = 9.0 Hz, 1H), 3.82 (dd, *J* = 9.0, 5.7 Hz, 1H), 3.67 (s, 3H), 3.44 (s, 3H). ^13^C NMR (100 MHz, CDCl_3_): *δ* 158.7, 154.5, 136.4, 135.0, 133.2, 132.6, 130.4, 130.2, 129.9, 128.8, 128.2, 126.6, 123.5, 113.6, 73.7, 59.0, 43.4, 29.2. HRESIMS calcd for [C_18_H_17_ClN_2_O_2_ + H]^+^ 329.1057, found 329.1077.

3-(2-Methoxyl-1-(naphthalen-2-yl)ethyl)-1-methylquinoxalin-2(1*H*)-one **4y**. Yellow solid (18 mg, 26%). Mp: 107–109 °C. ^1^H NMR (400 MHz, CDCl_3_): *δ* 8.00 (d, *J* = 8.0 Hz, 1H), 7.91 (s, 1H), 7.83–7.76 (m, 3H), 7.67 (d, *J* = 8.4 Hz, 1H), 7.54 (t, *J* = 8.0 Hz, 1H), 7.46–7.40 (s, 2H), 7.39 (d, *J* = 7.6 Hz, 1H), 7.27 (d, *J* = 8.4 Hz, 1H), 5.23 (t, *J* = 7.3 Hz, 1H), 4.49 (t, *J* = 9.0 Hz, 1H), 4.03 (t, *J* = 8.0 Hz, 1H), 3.63 (s, 3H), 3.43 (s, 3H). ^13^C NMR (100 MHz, CDCl_3_): *δ* 159.1, 154.5, 136.2, 133.5, 133.1, 132.7, 132.6, 130.3, 130.0, 128.2, 127.9, 127.6, 127.5, 126.9, 125.9, 125.6, 123.5, 113.6, 74.7, 59.1, 47.5, 29.2. HRESIMS calcd for [C_22_H_20_N_2_O_2_ + H]^+^ 345.1603, found 345.1615.

3-(2-ethoxy-1-phenylethyl)-1-methylquinoxalin-2(1*H*)-one **4z**. Yellow solid (38 mg, 62%). Mp: 72–74 °C. ^1^H NMR (400 MHz, CDCl_3_): *δ* 7.93 (dd, *J* = 8.0, 1.3 Hz, 1H), 7.51 (dt, *J* = 1.3, 8.5 Hz, 1H), 7.46 (d, *J* = 7.3 Hz, 2H), 7.34 (t, *J* = 1.0, 8.0 Hz, 1H), 7.28 (t, *J* = 7.3 Hz, 1H), 7.26 (t, *J* = 8.5 Hz, 2H), 7.20 (dt, *J* = 1.0, 7.3 Hz, 1H), 5.03 (dd, *J* = 8.8, 6.0 Hz, 1H), 4.44 (t, *J* = 9.2 Hz, 1H), 3.90 (dd, *J* = 9.5, 6.0 Hz, 1H), 3.62 (s, 3H), 3.56 (q, *J* = 7.0 Hz, 2H), 1.13 (t, *J* = 7.0 Hz, 3H). ^13^C NMR (100 MHz, CDCl_3_): *δ* 159.3, 154.6, 138.8, 133.1, 132.7, 130.2, 129.9, 128.7, 128.5, 127.0, 123.4, 113.5, 72.5, 66.5, 47.5, 29.1, 15.1. HRESIMS calcd for [C_19_H_20_N_2_O_2_ + H]^+^ 309.1603, found 309.1620.

## 4. Conclusions

In summary, we have developed a novel GO/visible light dual catalytic system for the multicomponent anti-Markovnikov alkoxylation of styrenes with excellent regioselectivity under mild reaction conditions. This methodology exploits a commercially available GO as the inexpensive and green photoredox catalyst for the synthesis of methoxylquinoxalones. In this study, various synthetically relevant functional groups, such as halides, trifluoromethyl, esters, allyl, and propargyl, can be compatible. We believe this GO/visible light dual catalytic strategy will add prospective luminescence over expensive and toxic metal-catalyzed reactions in organic synthesis.

## Data Availability

Data is contained within the article or Appendix A.

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
