# Peer review of "Photoinitiated Multicomponent Anti-Markovnikov Alkoxylation over Graphene Oxide"

_molecules, 2022, doi:10.3390/molecules27020475_

Round 1

Reviewer 1 Report

  • Specific comments to the authors

    The present manuscript illustrates photoredox catalysis in N-methylquinoxalone heterocycles preparation that employs multi-component synthetic procedure in anti-Markovnikov addition between alkoxyl radical and styrenes. The reported MCR reaction achieved by employing non-metal catalyst, i.e. graphene oxide, besides waste consumption should be reduced. The reported protocol has been provided broad functional group tolerance also. This is the foremost manuscript reported in graphene oxide catalyzed and visible-light-induced multicomponent anti-Markovnikov alkoxylation. Therefore, I recommend this manuscript after minor corrections.

    1)      In scheme 2, under air condition obtained trace amount of product 4a, trace means how much % of yield is obtained, should mention in the text.

    2)      In page 6, no reaction took place under an oxygen condition only not in the air, under air atmosphere authors were reported a trace amount of product achieved.

    3)      1H NMR data of the compound 4f is identified extra one proton signal, authors should check and report accordingly. 

Author Response

Response to reviewer 1:

  1. In scheme 2, under air condition obtained trace amount of product 4a, trace means how much % of yield is obtained, should mention in the text.

Reply: We agree with your suggestion and trace means <5% yield.

  1. In page 6, no reaction took place under an oxygen condition only not in the air, under air atmosphere authors were reported a trace amount of product achieved.

Reply: We agree with your suggestion and have done the correction.

  1. 1H NMR data of the compound 4f is identified extra one proton signal, authors should check and report accordingly. 

Reply: We agree with your suggestion and have done the correction.

Reviewer 2 Report

The Manuscript submitted by Peng, Huang, Liu and their co-workers describes a good, sometimes very good and even excellent as seducing, in some key points, Science. The main merit of the Authors consists of letting their own science speak by itself. My strong recommendation for the Editor “Accept after major revision” refers to rather formal (?) aspects that I attempt to resume below. In this purpose, I kindly ask the Editor to forward to the Authors the attached PDF-file containing a plethora of annotations marked in green (suggestions), yellow and red (criticisms, inquiries, unclarities etc).

  1. The English must be improved almost throughout the Manuscript, i.e., in no case should the Authors consider my incipient corrections as an “effort” to re-edit (re-write?) their work. The vocabulary is not rich enough compared to the level of science presented. Short sentences are, however, generally acceptable. On the contrary, the construction of (long) phrases denotes a poor knowledge of the logical relationship between the main sentence (subject) and its subordinates, attributive or complementary.
  2. Use of GO. If this catalyst was commercial as well (? lines 188, 189), its overall nature (structure? porous? membrane? see line 26) is not commented with respect to its reactivity (Schemes 1b and 3). Try to be more specific, please: is that representation “generic” or “illustrative”? Just by curiosity (of mine only?): could this type (?) of GO catalyst re-cycled?
  3. Quinoxalones. Were the type 1 starting materials commercial as well? Were they prepared for the current research? If not, published where? I did not find any dedicated literature in the References list. Finally: was there a particular interest in testing these substrates in their study? When putting this simple question, I made no connection with the Authors’ affiliation.
  4. Lines 416, 417: the SI section is incomplete.

Author Response

Response to reviewer 2:

  1. The English must be improvedalmost throughout the Manuscript, i.e., in no case should the Authors consider my incipient corrections as an “effort” to re-edit (re-write?) their work. The vocabulary is not rich enough compared to the level of science presented. Short sentences are, however, generally acceptable. On the contrary, the construction of (long) phrases denotes a poor knowledge of the logical relationship between the main sentence (subject) and its subordinates, attributive or complementary.

Reply: We agree with your suggestion and have done the correction. We also invited professional organizations to modify and polish our manuscript.

  1. Use of GO. If this catalyst was commercial as well (? lines 188, 189), its overall nature (structure? porous? membrane? see line 26) is not commented with respect to its reactivity (Schemes 1b and 3). Try to be more specific, please: is that representation “generic” or “illustrative”? Just by curiosity (of mine only?): could this type (?) of GO catalyst re-cycled?

Reply: We agree with your suggestion. “The GO material used in this investigation was prepared by Hummers oxidation of graphite and subsequent exfoliation, as reported. The obtained GO material has been characterized by X-ray photoelectron spectroscopy (XPS), scanning electron microscop (SEM), and infrared spectrum (IR) (see the Supporting Information).” was added in page 2. “The catalyst reusability was examined. The GO was separated by filtration after the first reaction run and used for the second one under the same conditions. However, the yield of 3a is very low, indicating that the recovered GO is ineffective.” was added in page 7. “X-ray photoelectron spectroscopy (XPS) and Fourier transform infrared (FT-IR) analysis showed that a large of epoxide groups and carbonyl functional groups are lost when GO participates the reaction (Figure S1 and S3). The GO and recovered-GO (GO-R) were analyzed by XPS in order to study the surface chemical state and chemical composition of GO and GO-R. The full-scale XPS spectrum (Figure S3) proved that GO have C, O and S elements. For GO-R, the presence of C, O, S, F and B were confirmed by survey XPS spectra, indicating that the anion BF4 was successfully doped in carbon catalyst.” was added in page 7.

  1. Quinoxalones.Were the type 1starting materials commercial as well? Were they prepared for the current research? If not, published where? I did not find any dedicated literature in the References list. Finally: was there a particular interest in testing these substrates in their study? When putting this simple question, I made no connection with the Authors’ affiliation.

Reply: We agree with your suggestion. The general procedure for the synthesis of quinoxalones 1 was added in SI page 3.

  1. Lines 416, 417: the SI section is incomplete.

Reply: We agree with your suggestion and have done the correction.

Round 2

Reviewer 2 Report

I am pleased to recommend the publication of the Manuscript submitted, in a convincing revised form, by Peng, Huang, Liu and their co-workers.